# Electrospun Alginate Nanofibers Toward Various Applications: A Review

**DOI:** 10.3390/ma13040934

**Published:** 2020-02-20

**Authors:** Teboho Clement Mokhena, Mokgaotsa Jonas Mochane, Asanda Mtibe, Maya Jacob John, Emmanuel Rotimi Sadiku, Jeremia Shale Sefadi

**Affiliations:** 1Department of Chemistry, Nelson Mandela University, Port Elizabeth 6031, South Africa; mjohn@csir.co.za; 2Advanced Polymer Composites, Centre of Nanostructured and Advanced Material, CSIR, Pretoria 0184, South Africa; amtibe@csir.co.za; 3Department of Life Sciences, Central University of Technology Free State, Private Bag X20539, Bloemfontein 9301, South Africa; mochane.jonas@gmail.com; 4School of Mechanical, Industrial & Aeronautical Engineering, University of the Witwatersrand, Johannesburg 2000, South Africa; 5Institute of NanoEngineering Research (INER), Department of Chemical, Metallurgical and Materials Engineering, Tshwane University of Technology, Pretoria 0001, South Africa; SadikuR@tut.ac.za; 6Department of Physical and Earth Sciences (PES), Sol Plaatje University, Kimberley 8301, South Africa

**Keywords:** electrospinning, alginate, nanofibers, tissue engineering, bioremediation, biofiltration, sensors

## Abstract

Alginate has been a material of choice for a spectrum of applications, ranging from metal adsorption to wound dressing. Electrospinning has added a new dimension to polymeric materials, including alginate, which can be processed to their nanosize levels in order to afford unique nanostructured materials with fascinating properties. The resulting nanostructured materials often feature high porosity, stability, permeability, and a large surface-to-volume ratio. In the present review, recent trends on electrospun alginate nanofibers from over the past 10 years toward advanced applications are discussed. The application of electrospun alginate nanofibers in various fields such as bioremediation, scaffolds for skin tissue engineering, drug delivery, and sensors are also elucidated.

## 1. Introduction

Resulting from their unique properties, such as their easy moldability, light weight, and relative inexpensiveness, polymeric materials play a significant role in our daily lives [1,2,3,4]. They have been employed in different fields, including drug delivery, catalysis, healthcare, and construction; hence, it is not possible to picture life without them. There is a welcome paradigm shift toward the use of polymeric materials that hold “green credentials”, regarding their distinctive properties, such as biodegradability, biocompatibility, renewability, and abundant availability [3,4]. The driving forces behind are associated with the rate at which our oil reserves are being depleted as well as the importance of reducing our carbon footprint. In this context, alginate merits special interest due to its unique features, *viz.*: high biocompatibility, fairly low immunogenicity, cheaper and excellent gel-forming capacity, as well as structural resemblance to proteoglycans (glycosaminoglycan (GAG)), which is a major component of the extracellular matrix (ECM) in human tissue. It has been one of the commonly employed biomaterials in various biomedical applications ranging from wound dressing and drug delivery to tissue engineering [5,6,7,8].

According to recent report from global forecast, the worldwide market for alginate is projected to reach USD 923.8 million by 2025 (https://www.prnewswire.com/news-releases/global-alginate-high-g-high-m-market-analysis-2014-2017--forecasts-to-2025---market-size-is-expected-to-reach-usd-9238-million-300587402.html). This is driven by its applications in food, textile, drugs, and clinical wound dressing. Alginate, also known as algin or alganic acid, is one of the natural polysaccharides found in the cell walls of brown algae [8,9,10,11]. It can also be produced by two bacteria genera, i.e., *Azotobacter* and *Pseudomonas* [9]. Alginate features unique properties, e.g., gelation, which allows its application in the textile, food, printing, and pharmaceutical industries. It is a linear copolymer composed of *β-d*-mannurate (M) and *α*-*l*-guluronate, which are linked through the 1–4 glycosidic bonds. It was first isolated from brown algae in 1880; however, its commercial production only began in the 20th century [8,12,13]. The properties of alginate directly depend on the sequence and composition of the M/G units. These units can vary depending on the source, growth conditions, maturity, season, and depth at which it is extracted. About 30,000 metric tons of alginates are industrially produced annually from brown algae (seaweed). It makes up 40% of the dry weight of the primarily used seaweeds, genera *Lamanaria* and *Macrocystis.* It is responsible for the flexibility and strength of the seaweeds, making it similar to cellulose in the plants. 

There has been unprecedented interest in the usage of the alginate toward different applications due to its unique valuable properties, such as its sustainability, biodegradability, biocompatibility, high hygroscopicity, flame retardancy, antimicrobial properties, and excellent ion adsorption [14,15,16]. It has been used in different forms, such as nanoparticles, microparticles, microspheres, and nanofibers, and it is widely applied in the food, textile, and biomedical industries as well as in bioremediation [17]. Beside the grafting of various functional groups, e.g., –SO_3_H and –NH_2_, grafting also offers another interesting property to alginate, i.e., exploring the electrospinning of alginates into ultrathin fibers, which enables a novel platform to fabricate highly complex materials [18]. The electrospinning technique serve as reliable processing method that is available to fashion different polymeric materials into nanofibrous materials, having a large surface area and malleable porosity. However, it is recognized that the electrospinnability (meaning the possibility of fabricating bead-free nanofibers) of alginate, similar to any other natural polymer, pose some difficulties [5,6,7,8]. This has been attributed to the several aspects, such as their polyelectrolytic nature, rigid intramolecular and intermolecular hydrogen network, lack of solubility, and high gelation at fairly low concentrations [5,6,7,8]. In order to overcome these issues, the electrospinnable polymers (e.g., polyethylene oxide (PEO) and polyvinyl alcohol (PVA)), strong polar solvents, and alginate modifications have been employed to facilitate its electrospinnability, as it will be discussed in the next sections. Furthermore, according to the modification of Nie et al. [19], the spinning of alginate into calcium chloride was found to produce nanofibrous membranes [19].

Besides the hurdles associated with the electrospinnability of alginate from its solution, there has been a lot of interest in the production of alginate nanofibers toward various applications. The literature database Web of Science returned with more than 340 references for the search query ‘electrospun alginate, electrospun alginic acid, electrospinning alginate, electrospinning alginic acid’, accessed on 25 November 2019. Figure 1 shows the year-wise number of references over the last 10 years. In this review, the recent trend in the exploration and exploitation of electrospun alginate nanofibers in various applications in the past decade are discussed. The trends on the electrospinnability of alginate are also illustrated. The main aim of this review in the light of the cited literature is to brief on the electrospinnability of alginate and their applications in bioremediation, scaffolds for skin tissue engineering, drug delivery, and sensors, which according to our knowledge has not been discussed before. It will also serve as a reference point for future research efforts with regard to much attention received by electrospun alginate nanofibers over the past decade.

## 2. Electrospinning Technique

Electrospinning has been a topical subject during the past two decades because of its capability to produce micro- or nanofibers from a wide variety of materials [20,21]. This technique, with its rich history, has been around for more than a century, as schematically summarized in Figure 2.

In this process, a voltage is applied to introduce an electric field between the syringe and the collector. The polymer droplet at the tip of the syringe, held by surface tension, is electrified in order to generate a jet, which is stretched and elongated to form fibers, having diameters ranging from a few nanometers to a few microns. The classic laboratory-scale electrospinning setup consists of four principal components, *viz.*: high-voltage supply, syringe pump, syringe, and ground collector, as shown in Figure 3. Basically, the pendant droplet, held by surface tension, is electrified and deforms into a conical shape (Taylor cone) by the repulsion of charges on the surface of the droplet and hence, an electrified jet is ejected. The electrified jet begins moving in a straight line, which is followed by a whipping instability. During the “whipping instability”, the electrified jet is elongated, stretched, and thinned into fibers, having nanosized diameters, which dry upon reaching the ground collector. These fibers feature distinctive properties, such as a large surface-to-volume ratio, interconnectivity structure, and malleable porosity. These properties can be fine-tuned by controlling the processing variables, such as solution properties (viscosity, conductivity, surface tension, concentration), setup parameters (voltage, tip-to-collector distance, collector shape, feeding rate, needle diameter), and ambient conditions (humidity and temperature) [22]. It is recognized that the solution properties are the most decisive parameters. The electrospinning begins from finding a suitable solvent to dissolve the polymer in order to afford sufficient solution viscosity (chain entanglements) and to avoid varicose breakup (to yield droplets) as the electrified jet moves toward the collector. Well-organized and detailed review papers, based on the effect of these parameters, are published in the literature [10,20,21,22,23,24].

Nonetheless, over the past decades, there have been significant advancements and modifications on the conventional laboratory-scale electrospinning technique (single-needle electrospinning), leading to the attainment of a high production rate, desired patterns, and morphologies [20,24]. This has resulted in numerous companies manufacturing nanofibers for different applications, including car filters, facial masks, and water filters. In addition, the size of the resulting fibers and the possibility of controllable structures have seen their applications in biomedical applications with a few being clinically approved [13]. 

## 3. Electrospinnability of Alginate

Alginate is composed of 1-4-linked *β-d*-mannuronic and *α-l*-guluronic acid residues, and it is obtained from the cell walls of brown algae (e.g., *Laminaria* and *Ascophyllum* species) [13,25]. Alginate, also known as alginic acid or algin, is a linear polysaccharide with the monomeric acid residues being covalently linked together in different compositions and sequences. Alginate acid residues are arranged in an “egg-box or worm-like” structure in the presence of the multivalent cations, as schematically shown in Figure 4. This structure confirmation and its polyelectrolytic character have been the main contributors for the limited electrospinnability of alginate solutions [14,26]. The polyelectrolytic tendency results in high electrical conductivity, which also contributes to the difficulties associated with alginate’s electrospinnability. In addition, the lack of chain entanglements, gelation at low concentration (i.e., below the formation of entanglements), and high surface tension also contribute to this limited spinnability [26]. Numerous researchers have been working on enhancing the spinnability of alginate through methods such as blending with hydrosoluble polymers, developing a co-solvent system, and modifying the alginate [25,27,28,29,30] (Table 1). 

### 3.1. Pure Alginate

Very few studies have reported on the electrospinnability of alginate solution without employing the carrier polymers [26,27,31]. Fang et al. [26] reported on the electrospinnability of pure sodium alginate. However, only irregular droplets were deposited onto the collector due to the lack of entanglements between molecular chains. They further introduced calcium chloride in order to enhance chain entanglements, which only led to non-continuous long fibers. It was reported that sodium alginate (SA) gelled by the presence of Ca^2+^ cations improved the molecular chain interaction, which facilitated alginate electrospinnability. Similarly, Saquing et al. obtained only droplets regardless of an increase in alginate content (1–6 wt %) [27]. It was indicated that the inability to electrospin alginate solution is not solely attributed to a lack of chain entanglements, but additional factors; e.g., surface tension and viscosity play major roles in the production defect-free alginate fibers. They found that the inclusion of non-ionic surfactant (Triton X-100) reduced the size of the droplets collected despite the twofold reduction of surface tension. In summary, the electrospinnability of alginate involves many aspects, such as solution viscosity, conductivity, surface tension, and its molecular structure. Gelation at fairly low concentration below the formation of the stable jet is also another aspect that contributes to this factor. Up until now, different strategies have been employed to improve alginate spinnability, as discussed in the next sections.

### 3.2. Co-Solvent Systems

Co-solvent systems involve the introduction of one or two additional solvents (usually less than 20 wt % overall weight than the one that dissolves the polymer) [19,32]. The fabrication of electrospun alginate nanofibers from co-solvent system solution was reported by Nie et al. (2008) [19]. The co-solvent system was composed of a strong polar glycerol and water, and this afforded nanofibers with an average diameter ranging from 120 to 300 nm when the concentration was increased from 1.6 to 2.4 wt %. The solution was directly electrospun into coagulation bath containing ethanol and 10 wt % CaCℓ_2_ aqueous solution for fiber crosslinking. The solution electrospinnability was attributed to glycerol, improving the chain entanglements of alginate and flexibility. In addition, the solution viscosity was enhanced, while both the surface tension and conductivity were reduced. Elsewhere, it was reported that the co-solvent system improved the spinnability of alginate by reducing the surface tension and electrical conductivity [26]. In this case, a complex solvent system composed of water, ethanol, and dimethylformamide (75/15/10) with calcium chloride 1.5 wt % resulted in a continuous fluid jet. 

### 3.3. Hydrosoluble Polymers

The use of hydrosoluble polymers (e.g., polyvinyl alcohol (PVA), polyethylene oxide (PEO), etc.) as carrier materials to facilitate alginate spinnability has been the most convenient process to fabricate bead-free fibers [12,25,27,28,29,31,32,33]. It was reported that the carrier polymers coordinate with the alginate polymer through hydrogen bonding, thereby reducing the strong intra- and inter-molecular network between the alginate chains. This results in reducing the viscosity, surface tension, and conductivity of the alginate solution, and hence enhancing the electrospinnability. The main drawback, besides the post-treatment for removing the ‘carrier’ polymer, is the limited amount of alginate that can be included into the blend (approximately 40%). In addition, the possibility of residual hydrosoluble polymers are often found within electrospun nanofibers after post-treatment [31]. Saquing et al. (2013) studied, thoroughly, the effect of the carrier polymer on the electrospinnability of alginate with regard to entanglements and viscosity [27]. The authors used PEO of different molecular weights and surfactant in order to obtain an insight on the influence of entanglements and viscosity. It was found that the hydrosoluble polymers coordinate with metal cations, such that the interaction between PEO and PEO is a decisive parameter for the electrospinnability of the blend. Three-dimensional electrospun alginate nanofibers were prepared for the first time by Bonino et al. [34]. The authors reported that the dissociation of carboxylic acid groups into negatively charged ions, induced Coulumbic repulsions between the fibers, thereby resulting in the fibers protruding toward the needle. This results in a three-dimensional structure formation. 

### 3.4. Co-Solvent/Surfactant and Carrier Polymers

Another suitable route for the electrospinning alginate involves the use of the co-solvent system and surfactant with the aim of increasing the content of alginate in the electrospinnable blend [28,35,36,37,38,39,40,41]. The presence of these components reduces the solution surface tension, as well as the conductivity and rigidity of the alginate, thereby resulting in bead-free fibers. Battarai et al. (2006) reported on the use of surfactant and co-solvent in order to increase the content of the alginate [31]. Bead-free nanofibers (diameter ranging between 40 and 100 nm) were obtained from 70:30 to 90:10 alginate/PEO ratios by employing 0.5 wt % Triton (non-ionic surfactant) and 5 wt % DMSO (co-solvent). Elsewhere, 85 wt % of alginate was electrospun in the presence of the non-ionic surfactant (Triton X-100) and carrier polymer (PEO). It was postulated that the surfactant further decreased the surface tension, which allows an increase in the alginate content in the blend [27]. Besides the use of the co-solvent system and carrier polymers in the presence of the surfactant, it was also reported that the solution storage over time can improve the electrospinnability of alginate solutions [28,35]. It was found that a storage time of 10 days at approximately 25 C is the optimal duration to afford cylindrical nanofibers having diameters ranging between 109 and 209 nm. It was postulated that during aging, the dissociated sodium ions (from alginate) and alginate’s carboxylate groups interact with PEO via ether oxygen, thereby improving the electrospinnability. In summary, the interaction between the carrier polymers with alginate is one of the principal factors that established the spinnability of the system. It is recognized that the spinnability of the carrier polymer is of significant importance because once it can be electrospun alone, then it is possible to spin its blend with alginate. Beside the incorporation of the co-solvent and surfactant, offering an increase in alginate content, they found that they (i.e., co-solvent and surfactant) enhanced the fibrous structure of the resulting scaffolds. This is often attributed to their reduction of surface tension, which serves as compensation for an increase in the electrical conductivity and a reduction in viscosity at higher alginate loading. The effect of surfactant on the three-dimensional structure alginate nanofibers was reported by Bonino et al. [34]. It was found that the alginate charges are concentrated on the interface, leading to repulsion between the neighboring fibers, which promotes the formation of 3D network of fibers. This phenomenon was not influenced by the presence of the surfactant. It was reported that the changes in humidity influence the 3D network structure due to its effect on the number of surface charges. A 30% relative humidity (RH) resulted in a 3D structure with bead-free fibers, having diameters of 237 ± 33 nm. This was attributed to more water being retained, thereby giving the fibers more surface charges and hence repulsion among them.

### 3.5. Structural Modification

The replacement of the co-solvent and surfactant can be achieved by modification of the alginate in order to replace the –OH groups, either via esterification, oxidation, or sulfation [7,21,42,43,44,45,46]. These processes introduce other functional groups onto the backbone of alginate by replacing the –OH groups, thereby reducing the intra- and inter-molecular hydrogen bonding network, which contributes to the rigidity in chain conformation [18,42]. Graft polymerization was applied in order to introduce the hydrophobic long-chains acrylonitrile (AN) on sodium alginate (SA-*g*-AN), such that the hydrogen bonding density decreases [14]. This is as a result of the hydroxyl groups being substituted by hydrophobic moieties of acrylonitrile. This process also reduces the surface tension and the electrical conductivity while enhancing the flexibility of the molecular chains. In addition, PEG was also added into the mixture (SA-*grafted*-AN) to facilitate the spinnability of alginate/polyacrylonitrile solution. SA-*g*-AN exhibited excellent electrospinnability with nanofibers that had pores on their surfaces, which became uniformly distributed as the concentration of acrylonitrile content increased. The presence PEG further improved the spinnability of the solution with an increase in PEG content and resulted in the improvement of the morphology from bead and spindled structure to bead-free nanofibers. Modaress and co-workers [43] used chlorosulfonic acid to introduce sulfate groups into alginate acid residues by substituting their hydroxyl groups. The authors managed to electrospin 50 wt % of sulfated alginate by using PVA as the carrier polymer and obtained bead-free fibers with an average diameter of 222 ± 62 nm. In addition, the sulfation reaction of alginate can be performed by employing other agents such as sulfuric acid, sulfuryl chloride, sulfamic acid, and sulfur trioxide [44,45]. In this regard, the presence of the sulfates groups along the alginate chains enhances its blood compatibility, cell adhesion, and anti-coagulating properties [44]. The fabrication of ammonium alginate derivatives was demonstrated by Pegg et al. [47]. It was reported that the amine-containing cargo—analgesics, antibiotics and therapeutic enzymes—can be introduced via ionic linkage and electrospun for drug delivery systems. Lidocaine (anesthetic), neomycin (carbohydrate antibiotic), and papain (a cysteine protease) were ionically linked to alginate and electrospun with the aid of PVA as the carrier polymer under the following conditions: 20 kV voltage, tip-to-collector distance (TOC) of 15 cm, and flow rate of 5 µL/min. They obtained fibers with diameters ranging between 100 and 300 nm for all the investigated systems. The authors further introduced all three cargos and concluded that there were no structural changes on the resulting electrospun nanofibers.

### 3.6. Advanced Electrospinning Techniques

The usage of other natural polymers to form a polyelectrolyte complex toward biomedical applications has also been reported in the literature [48,49,50]. The polyelectrolyte complex made up of chitosan and alginate was prepared by employing a co-axial electrospinning technique [48]. In this regard, the carrier polymer (PEO) was used to facilitate the spinnability of both chitosan and alginate. The optimal conditions were found to be as follows: voltage of 27 kV, distance of 10 cm, and feeding rate of 0.4 mL/h in order to afford defect-free fibers, having an average diameter of 154 ± 35 nm. It was reported that the co-axial nanofibers were obtained from PEO (1.6%)/SA (2.4%) (1:1 *w*/*w*) as the core and chitosan (5%)/PEO (4%) (70:30 *w*/*w*) as the shell. In another study, a complexation of alginate and chitosan was prepared by using the dual jet electrospinning system [51]. The combination of alginate and other polymers was found to improve the cell adhesion, chemical stability, and mechanical properties of the resulting blended composite material [25,49,51,52,53]. Elsewhere, alginate and polycaprolactone (PCL) were co-blended by the dual jet electrospinning system of their mixture with PEO onto a mandrel collector in order to facilitate cell adhesion onto the nanofibers [52,54]. It was demonstrated that the fibers, having different ratios (alginate: PCL), can be achieved by changing the feeding rate of the polymer jets.

The modification of the collector in order to facilitate the fabrication of the electrospun nanofibrous alginate was also reported [19,40,55,56]. In this regard, the electrically grounded collector is immersed in bi/multivalent ion aqueous solution for simultaneous crosslinking of the collected alginate nanofibers. Furthermore, the carrier polymer is also removed from the fibers. Vicini et al. (2018) prepared nanofibrous membranes from PEO/alginate and hyaluronic acid (HA)/PEO/alginate solutions by directly spinning onto a collector immersed in a solution of barium in 40:60 ethanol/water. The removal of PEO from the fibers was confirmed by differential scanning calorimetry (DSC) analyses. It was reported that after crosslinking with barium collecting solution, the melting peak of PEO (i.e., at about 60 °C) disappeared. The obtained tissue-like mats were found to consist of ribbon-like fibers (up to 5–6 μm) due to the fusion of nanofibers for the ternary system (Alginate/PEO/HA). Comparison between the conventional electrically grounded and collector immersed in a solution was conducted by Castellano and co-workers [56]. It was found that the conventional collector led to a collapsed structure having a less porous membrane with fiber diameter of 150 ± 30 nm, while the solution-immersed collector resulted in a highly porous structure with fiber diameters of 100 ± 30 nm. In this case, thermogravimetric analysis (TGA) was used to verify the removal of PEO from the electrospun nanofibers. Beside the crosslinking resulting in highly thermal stable materials as compared to uncrosslinked alginate, it was reported that the degradation peak associated with PEO disappeared, indicating the successful removal of PEO as the carrier polymer.

### 3.7. Other Polymers

Other polymers were also used as carrier polymers for facilitating the spinnability of alginate [57,58]. Polylactic acid (PLA) as the continuous phase and sodium alginate as the dispersion phase were used by Xu et al. [59] to prepare a W/O emulsion and then electrospun into nanofibrous membrane for tissue engineering. The authors separately dissolved sodium alginate in water and PLA in chloroform and then mixed the solutions together in order to yield emulsions. The electrospinning conditions were kept as follows: a flow rate of 0.5 mL/hr, voltage of 15 kV, and distance to collector of 15 cm in order to obtain ultrafine fibers having diameters of about 250 ± 90 nm. It was found that alginate was dispersed along the fibers with no difference with regard to the alginate content in the blend. The resulting blend exhibited excellent mechanical properties with tensile strength increasing from 0.25 to 3.13 MPa. Elsewhere, electrospun nanofibers composed of poly (lactic-*co*-glycolic acid) (PGLA) and ciprofloxacin-loaded alginate were fabricated by using the classical single needle electrospinning technique [58]. In this regard, alginate was spray dried in order to produce particles with sizes ranging between 100 nm and 15 µm and suspended into chloroform; then, it was introduced into PLGA–trifluoroethanol solution. It was reported that bead-free fibers were obtained (diameters of between 633 and 877 nm), even after the introduction of ciprofloxacin into the system under investigation.

## 4. Applications of Electrospun Alginate

Electrospun alginate nanofibers have seen their application in various fields, as summarized in Table 1. Of all the proposed applications, the biomedical field has been the most explored or exploited due to the structural similarity to the ECM, as recently reviewed by Taemeh et al. [8]. However, few studies showed that electrospun alginate nanofibers can be employed to other fields, including bioremediation, as will be discussed in subsequent sections. The limitations in its applications result from alginate solubility in water and a lack of electrospinnability from its aqueous solutions.

### 4.1. Wastewater Treatment

#### 4.1.1. Bioremediation

Electrospun alginate nanofibers have been used to prepare nanofibrous membranes for metal adsorption [10,64,65]. The high porosity and large surface area-to-volume ratio of alginate electrospun were fabricated by using polyethylene oxide as the carrier polymer for copper (as model metalloid) adsorption, as reported by Mokhena et al. (2017) [28]. A macroporous fibrous membrane (with an average pore size of 215 nm, surface area of 18.03 m^2^g^−1^, and average pore volume of 0.96 cm^−3^g^−1^) and the presence of hydroxyl and carboxyl groups facilitated the removal of copper ions from aqueous medium (i.e., reaching maximum adsorption capacity of 15.6 mg g^−1^). It was shown that the membrane can be reused five times by soaking it in ethylenediaminetetraacetic acid (EDTA) solution without deterioration of its absorbance capabilities. In addition, the membrane exhibited excellent selectivity toward copper and nickel ions; hence, it can be used for their recovery. In another study, the authors used electrospun alginate nanofibers as a substrate, coated with cellulose nanocrystals as a barrier layer for chromium ((Cr(VI)) removal [35]. The presence of sulfates and hydroxyl groups from cellulose as well as hydroxyl and carboxyl groups from alginate enhanced the removal at high pH values to more than 70%.

The removal of dyes from water streams has been one of difficult processes due to their inertness [66]. Electrospun alginate nanofibers with active carboxyl and hydroxyl groups have been explored for the adsorption of dyes from wastewater streams [59]. In addition, electrospun mats render additional features, such as their porosity, tunable pore size, and large surface area, hence providing promising material for water purification. Since electrospun alginate nanofibers are hydrosoluble, crosslinking is often applied to improve their stability in water. A recent study by Tan and co-workers demonstrated that the crosslinking of alginate nanofibers by using different agents resulted in different adsorption capacities for methylene blue dye [59]. The authors crosslinked alginate nanofibers by using classic calcium chloride (CaCℓ_2_), glutaraldehyde vapor (GA), and trifluoroacetic acid (TFA) and found that the membranes exhibited excellent adsorption capacity, which was maintained after five cycles (>90%). The maximum adsorption capacities were 2357.87, 1755.23, and 1863.10 mg/g for CaCℓ_2_, GA vapor, and TFA crosslinked alginate mats, respectively. In addition, the mats exhibited different adsorption efficiencies in various media, indicating that they can be used in different environments, depending on the crosslinking agent employed. 

#### 4.1.2. Filtration Membranes

Since the wettability properties of the materials play a major role in the oil-in-water separation, the modification of the inherently hydrophilic alginate polymer into special wettable surfaces, *viz*., superhydrophobicity (low water affinity), superhydrophilicity (high water affinity), superoleophobocity (low oil affinity), or superoleophilicity (high oil affinity). Yu and co-workers [14] succeeded in changing the surface properties of alginate by electrospinning the alginate-grafted-acrylonitrile solution for oily water purification. They obtained highly hydrophobic nanofibers, which improved when the content of AN increased (i.e., water contact angle increased from 56° to 70°). This was attributed to the presence of the strong polar CN groups, which increased the interaction of the molecular chains in the system. The hydrophilic nature of calcium crosslinked electrospun alginate was explored for oil retention by Luyt and co-workers [35]. It was found that the alginate-based membrane exhibited excellent a retention level of 96.6%. The coating of cellulose nanowhiskers improved the integrity of the membrane and enhanced the retention capacity to 98.4%. Similarly, coating with chitosan or chitosan/silver nanoparticles (AgNPs) onto electrospun alginate nanofibers enhanced the oil retention from 33% to approximately 94% [37]. This was ascribed to the formation of compact structure by coating these materials and thereby reducing the surface porosity of alginate nanofibers and hence improving oil removal.

The filtration membranes for dye separation were also reported [66,67]. Guo et al. [67] prepared a composite membrane that consisted of three layers: polyhydroxybutyrate (PHB) as substrate, PHB–calcium alginate nanofibers as a mid layer, and calcium alginate multi-walled carbon nanotubes as a selective layer. The membrane exhibited high flux of approximately 32.95 L/m^2^hr and a rejection rate of approximately 98.20%. The removal of Congo red (CR) dye by the adsorption process onto electrospun alginate crosslinked with glutaraldehyde was reported by Mokhena and Luyt [37]. They reported a maximum removal of more than 95%, which was associated with ionic interaction between the surface carboxyl groups and amino groups in the CR as well as the large surface area. The maintenance of high dye removal (i.e., >95%) over five cycles was achieved by coating with chitosan/AgNPs onto alginate nanofibers, which reinforced the alginate. This resulted in a compact structure and delayed the damage of the membrane under high filtration pressures.

### 4.2. Biomedical Applications

The advancements in the medicine field clearly indicate that in the near future, it may rely solely on the early detection of disease and prevention before its manifestation. The fabrication of multifunctional nanofibrous materials that have different chemical, physical, and mechanical properties by incorporating other components offers the opportunity to develop advanced products for medical applications [49,52]. Due to its low cost, low toxicity, biocompatibility, and biodegradability, alginate has been extensively studied for biomedical application [58,62,68].

#### 4.2.1. Wound Dressing

Wound healing is considered a complex process that involves a series of events, such as cell response, growth, and differentiation as well as a healing microenvironment and patient afflictions (age and illness) [69]. Therefore, the wound-care products must possess durability, non-toxicity, and flexiblility, while also being non-antigenic to facilitate wound repair. Besides the attractive attributes of alginate (*viz*., biocompatibility, biodegradability, and non-toxicity), electrospinning renders alginate additional structural features for wound healing such as acceptable porosity to assist in the transport of oxygen and moisture regulation onto the affected site while keeping bacteria out and malleable mechanical properties comparable to human skin. Moreover, bioactive substances can be incorporated to produce multifunctional products that can further promote wound healing [70]. PLGA/sodium alginate (SA) loaded with ciprofloxacin (antibiotic) for wound dressing was prepared by Liu et al. [58]. Sodium alginate enhanced the hydrophilicity of the samples from approximately 111.7° to 97.3° and with excellent mechanical properties with its Young’s modulus being close to that of human skin (i.e., tensile strength and modulus of about 60–70 MPa and 17–21 MPa, respectively). With the acceptable drug-release rate and antibacterial activity exhibited by these nanofibrous membranes, they can be used for wound dressing. Beside the antimicrobial activity of the electrospun mats, the water vapor transmission rate (WVTR) is of significance in supporting wound healing. The antimicrobial alginate mats, coated with chitosan/silver nanoparticles, exhibited good WVTR values that are acceptable for wound dressing, *viz*. 1373–1586 gm^−2^day^−1^ [36]. It recognized that sodium alginate is soluble in water, which requires post-treatment in order for it to be suitable for the encapsulation of antibiotic drug into electrospun nanofibers. In most cases, the utilization of ethanol/CaCℓ_2_ post-treatment may remove the loaded antibiotic drug for wound dressing (which is often loaded into an electrospinnable blend) [39]. In addition, the burst release of the drug at the initial state is often observed [39]. In this case, the crosslinking of the carrier polymer, rather than alginate, has been proposed as a suitable method to overcome such hurdles. Yang et al. [21] used glutaraldehyde as the crosslinking agent to fabricate hydrogel from electrospun PVA/SA. A thermally stable hydrogel was obtained after crosslinking, with acceptable cell growth and viability for tissue engineering applications. Residual solvent during electrospinning can also be conducive for crosslinking the electrospun nanofibers, as demonstrated by Wei et al. [12]. By exploring the high water-binding capacity of alginate, the authors found that the residual solvent led to the crosslinking of the fibers to a three-dimensional structure. The presence of the silicate layers and the hydrophilic character of alginate improved the antibacterial activity against *E. coli* and *S. aureus*, which opens the door for their applications in wound dressing. Dodero et al. [71] also reported on the crosslinking of alginate nanofibers with strontium in the presence of zinc nanoparticles (ZnNPs) for improving the antibacterial property of the ensuing membrane. The resulting nanofibrous membrane had mechanical properties similar to human skin (*viz*., tensile modulus and strength of 280–470 MPa and 95–160 MPa) and acceptable water vapor (3.8−4.7 × 10^−12^) for wound-dressing mats; it exhibited excellent antibacterial activity with less cell adhesion, and hence can be used for wound dressing and surgical patches. 

#### 4.2.2. Tissue Engineering

Cell adhesion and viability on the electrospun alginate nanofibers has been one of the major limitations for their applications in tissue engineering. Composite materials composed with bioactive substances to promote cell adhesion and viability has been one of the suitable routes for the application of electrospun alginate nanofibers in tissue engineering [72]. Jeong et al. [5] prepared a covalently-bonded cell adhesive peptide (glycine-arginine-glycine-aspartic acid-serine-proline (GRGDSP)) to electrospun alginate in order to enhance cell viability and growth on the fibers. The presence of the adhesive peptide did not influence the morphology of the resulting fibers, but meanwhile, it enhanced cell adhesion and spreading, even after 4 h of post-seeding. This was attributed to the nanoscale fibrous structure and adhesive peptide. Similarly, crosslinked alginate nanofibrous mats were first soaked in 1-ethyl-3-(3-dimethylaminopropyl) carbodiimide (EDAC) solution and then immersed overnight in collagen in order to enhance cell adhesion and proliferation onto fibers [6]. It was reported that the fibroblast cells showed good cell adhesion and proliferation due to the presence of amino acids from the collagen. The nanofibrous structure and hydrophilic nature of the mats initiated the release of biological signal for cell adhesion and spreading on the scaffold. In another study, Jeong et al. [73] electrospun a polyionic complex of alginate and chitosan in order to realize cell attachment and proliferation. An increase in chitosan content decreased the swelling ratio in deionized medium and the seeded mouse preosteoblast cells (MC3T3s) were found to adhere to the chitosan–alginate nanofibrous membrane and exhibited substantial proliferation from 5 to 120 h. The incorporation of nanoparticles into alginate nanofibrous membrane resulted in the enhancement of the mechanical properties and usefulness as multifunctional materials for tissue engineering applications [29,33,61,74]. Antiseptic (cephalexin (CEF))-loaded halloysite nanotubes (HNTs) were incorporated into an alginate-based nanofibrous membrane by mixing the HNT-loaded particles in an alginate/PVA solution, followed by electrospinning [74]. The electrospun nanofibrous composite membrane was crosslinked by using glutaraldehyde. The membrane displayed good mechanical properties with tensile strength values of between 1 and 3.8 MPa (which is well above tissue engineering requirements) and excellent antibacterial activity against many bacteria (i.e., *S. aureus, P. Aeruginosa, S. epidermis, E. coli*) at only 10% *w*/*w* of CE-HNT, as shown in Figure 5.

In situ synthesis of hydroxyapatite (HAp) onto alginate nanofibrous mats during crosslinking afforded the homogeneous distribution of deposition of HAp nanocrystals along the fibers for bone tissue engineering [75]. In this case, the HAp nanocrystals precipitated and grew along alginate fibers, serving as a template for in vitro biomineralization with time. It was found that the rat calvarial osteoblast (RCO) cells were attached to the surface of the membrane and flattened, stretched, and elongated into a spindle-like structure, indicating the cell growth, proliferation, differentiation, and migration. In a study conducted by Hajiali et al. (2015), it was demonstrated that the degradation of alginate nanofibers can be adjusted to afford their application in regenerative medicine [63]. The authors used trifluoroacetic acid (TFA) at different intervals (3–24 h) to acidify carboxylate groups in order to generate very stable poly (alginic acid) (*viz*., degradability extended from 7 to 14 days in phosphate buffer solution (PBS) without damaging the nanofibrous structure of the membrane. The obtained membrane was biocompatible, with fibroblast cells (MTT assay) being able to attach on the fibers, indicating their potential in the fabrication of biomedical devices for regenerative medicine.

The easy preparation of artificial nerve grafts with structural and mechanical properties similar to a native extracellular matrix (ECM) for a conducive microenvironment for neotissues has been achieved using the electrospinning technique by Golafshan et al. [76]. In this case, a hybrid of graphene–PVA–alginate scaffolds was prepared in order to improve the mechanical property, degradation stability, and conductivity of the resulting mats, while maintaining the architecture to support the cell adhesion, growth, and cytokinesis. It was reported that the resulting scaffold displayed excellent mechanical properties (tensile strength: 22.1 ± 2.1MPa and toughness: 2.1 ± 1.2 MPa) suitable for neurotic engineering. The cultured PC12 cells were found to be tightly bound to the scaffold with rounded morphology on day 1; and proliferation of the cells (using MTT assay) statistically enhanced from 113.2 ± 2.12 (% control) on day 1 to 244.4 ± 10.3 (% control) for day 7, indicating that the resulting scaffold can be used for nerve tissue engineering.

#### 4.2.3. Cancer Therapy

Recent study on the magnetic alginate-based electrospun mats showed the potential of using these materials for cancer therapy [77]. In this case, the alginate chelates the Fe^2+^ ions used for Fe_3_O_4_ formation. In addition, these ions can crosslink alginate via ionic crosslinking, but the authors further crosslinked the mats by using glutaraldehyde, which extended their stability in water during a 72 h immersion. They conducted in vitro hyperthermia experiments on these mats and concluded that these mats can successfully be used for tumor treatment by either endoscopic delivery or as supplementary patch after surgical removal of tumor. Post-operative peritoneal adhesions can cause healthcare difficulties, since they lead to infertility, abdominal pain, and reoperations [78,79]. The current available membranes (e.g., Seprafilm^TM^, Interceed^TM^, etc.) are reported to be sticky, degrade quickly, and cause leukocyte response. It was recently reported that the blend of chitosan and alginate can be used for anti-adhesion products because of the unique properties of these materials [78]. The main aim is to exploit the lack of cell adhesion on the alginate-based materials and chitosan for inhibiting cell growth and triggering apoptosis. It was found that alginate reduced cell attachment and protein adsorption, while chitosan inhibited cell growth and triggered apoptosis. The blending of these polymers also improved their stability in Dulbecco-modified Eagle medium (DMEM) solution; meanwhile, electrospun mats offer flexibility and cause less stimulus response. 

#### 4.2.4. Delivery Systems

Beside regenerative gene therapy being the most reliable process to promote tissue repair, the difficulties associated with the delivery of these therapeutic genes makes it complicated. In order to overcome these challenges, these therapeutic genes are often immobilized on the biomaterial-based substrate. This kind of process promotes delivery efficiency on the targeted sites without systemic infections and/or immune responses. Electrospun nanofibers as a template to encapsulate or immobilize the vectors offers a novel route to control their release rate toward targeted sites, as recently reviewed by [80]. Hu and co-workers from the National Central University electrospun alginate with the aid of PEO (while adding PCL to overcome the limited cell adhesion for alginate-based materials) for gene delivery systems [53,54,58]. The immobilization of the genes was achieved by exploiting anionic character of alginate to form a polycomplex with cationic nonviral vectors e.g., polyethyleneimine (PEI) and chitosan via electrostatic interaction. In this case, a cationic PEI/DNA nanocomplex was prepared and stabilized on electrospun nanofibers for transfection in the presence of PCL. It was reported that the PCL promoted the cell viability while the presence of alginate enhanced the in situ transfection ability of the multifunctional composite material, as shown in Figure 6. The levels of transgene expression indicated that the higher alginate ratios resulted in the more transfected cells [58]. In another study, it was demonstrated that the immobilization of the genes can be enhanced by employing a direct-current electric field (DCEF) in order to guide the cationic polyplexes toward nanofibers on cathode [81]. It was reported that DCEF treatment reduced the adsorption time to 30 min to obtain the same level as for passive treatment. Furthermore, the adsorption after 2 h was found to be two times higher than one from passive treatment, indicating that the applied electric field is efficient for accelerating gene immobilization; however, this comes at the expense of alginate degradation. Nonetheless, the degradation of alginate due to DCEF treatment facilitated cell growth. The cell growth of DCEF-treated nanofibers showed cell extension and infiltration, confirming the improved biocompatibility of the mats, while facilitating the gene immobilization. 

Electrospun alginate nanofibers can also be used for enzyme immobilization toward various applications, e.g., biocatalysts. Teke and co-workers immobilized lipase onto alginate in order to enhance their stability [82]. It was found that the amount of the immobilized enzymes depended on the component of the blend (*viz*., 1.0 mg/mL for PEO/alginate (4.78 U/mg protein) and PVA/alginate 1.5 mg/mL (14.68 U/mg protein). The membranes showed excellent re-usability of 14 and 7 times for PEO/alginate and PVA/alginate, respectively.

### 4.3. Sensors and Energy

Electrospun nanofibers provide new platform for the design of novel sensors with high sensitivity, selectivity, capability, and portability [10,83]. This is a result of their unique features, which include their large surface-to-area ratio, malleable mechanical properties, and the ease of surface functionality modification. Different sensing agents can be embedded into electrospun nanofibers, thereby enhancing the sensitivity, response time, and detection level (lowest detection concentration) [10,83]. Alginate having carboxyl and hydroxyl groups can be functionalized with heavy metal-sensitive compounds, since it can bind to the multivalent ions and thus can be detected easily [10,84]. In this case, the fibers can be labeled with highly selective fluorescent sensors (by exploiting –OH and –COOH) for detecting metal ions in aqueous solution with ultralow detection limit (in nM) due to their large surface area [84]. Zhang et al. [85] exploited the rich carboxyl groups alginate to disperse silver (Ag) ions over the entire electrospun alginate nanofibers for humidity-responsive materials toward breath sensors to monitor breathing during exercising and emotion changes. In this case, Ag ions were introduced onto fibers via ion exchange followed by in situ Ag nanoparticles (AgNPs) throughout the fibers as shown in Figure 7. By attaching the obtained membrane onto the exhaling valve gasket of the 3M-9001V mask, it was demonstrated that the membrane was able to sense the breathing rate either during running (13 times per minute) or normal condition (16 times per minute) by measuring respiration frequencies. The mask yielded the same results after 3 months, indicating its stability and reusability. In the case of emotion changes, it was demonstrated that the membrane was capable of distinguishing the sadness and delight by monitoring breathing frequencies (i.e., breath frequencies reached value of 8 and 14 times per minute for sadness and delight, respectively). Furthermore, the membranes exhibited good properties to predict the breathing changes during sleeping in order to give an alarm for sleep apnea.

A similar strategy of ion exchange from sodium to Ag ions was also used to fabricate pressure sensors by Hu et al. [86]. The obtained nanofibers were integrated with planer patterned polydimethylsiloxane (PDMS) films to fabricate a wearable pressure sensor. The constructed pressure sensor exhibited excellent durability and reusability (>1000 cycles) with an ultralow detection limit (1 Pa), which can be applied in wearable artificial electronic skin and smart textile for the real-time detection of human activity signals.

On the other hand, electrospun alginate nanofibers, as an excellent battery electrode, can be explored in energy storage materials [83,87,88]. Similarly, a large number of carboxyl groups along alginate chains offers the sites for accommodating the nanoparticles for battery electrode, thereby enhancing the overall performance of the battery, as demonstrated by Kavalenko et al. [82]. It was reported that the inclusion of the alginate improved the stability of the anode, which delivered a reversible capacity eight times higher than the conventional graphitic-based anode.

## 5. Future Trends and Conclusions

From the works presented in this review, it can be concluded that electrospinning renders alginate a new platform toward different applications because of the unique properties of the resulting fibers. Calcium chloride is the commonly used crosslinking agent for the crosslinking of electrospun alginate nanofibers without losing their nanofibrous structure. The use of other crosslinking agents offers some degree of stability of alginate in different media, which sees alginate being used in other applications (e.g., bioremediation), other than the typical biomedical application. With modification of the alginate structure, there is a possibility of new applications, e.g., stimulus-responsive material, especially for sensing metalloids from wastewater streams, being discovered in the near future. The inclusion of nanoparticles offers a novel platform for fashioning multifunctional electrospun alginate-based materials toward various applications. It can also be concluded that it is still impossible to electrospin alginate aqueous solution without the aid of co-solvent and/or copolymer. The content of alginate in an electrospinnable blend can be increased by either modifying the alginate or employing surfactant and co-solvent. In the future, we foresee the derivatives of alginate being electrospun without the aid of carrier polymers and/or co-solvent. Beside PEO and PVA being the most commonly used carrier polymers, PVA renders an opportunity for crosslinking by using various crosslinking agent to improve the stability of the ensuing membrane. This is of interest with regard to the solubility of alginate in water or other media, which limits its application to other fields, aside from biomedical. In addition, PVA offers an opportunity to add other components (e.g., antibiotic, nanoparticles, biocides, etc.) into electrospinnable composition. The use of cationic biopolymers to form a polyionic complex is the most suitable method for crosslinking alginate nanofibers without their ‘green’ credentials being compromised. This method increases the stability of these fibers in different media; however, the complications during electrospinning such materials makes them more difficult to produce. Future advancements in electrospinning techniques may just be the suitable hope for producing polyionic materials; however, such techniques may be pricy for some institutions/research laboratories. Furthermore, most of the conducted research is only applicable at the laboratory scale; hence, the investigations on the industrial scale as well as real-life scenarios are of significant in order to be conclusive on how far these electrospun alginate nanofibrous materials are from reaching the market.

## Figures and Tables

**Figure 1 materials-13-00934-f001:**
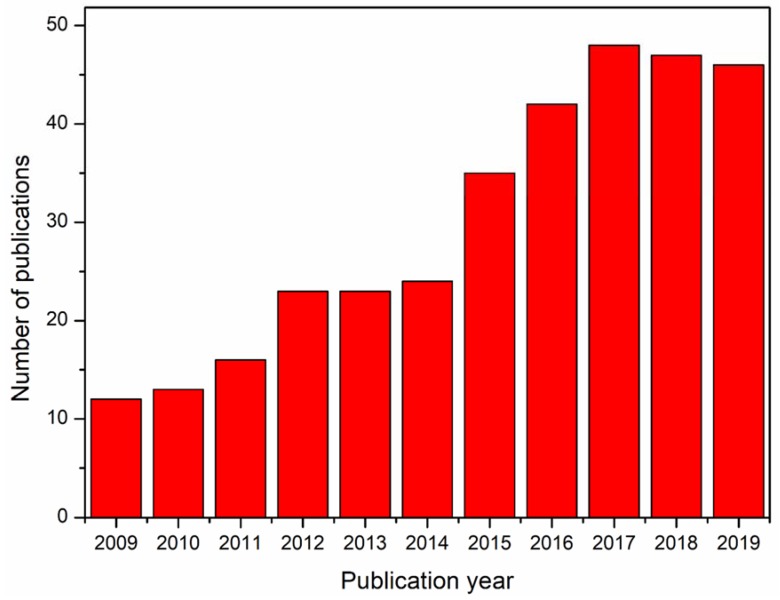
Illustration of the annual number of scientific publications since 2009.

**Figure 2 materials-13-00934-f002:**
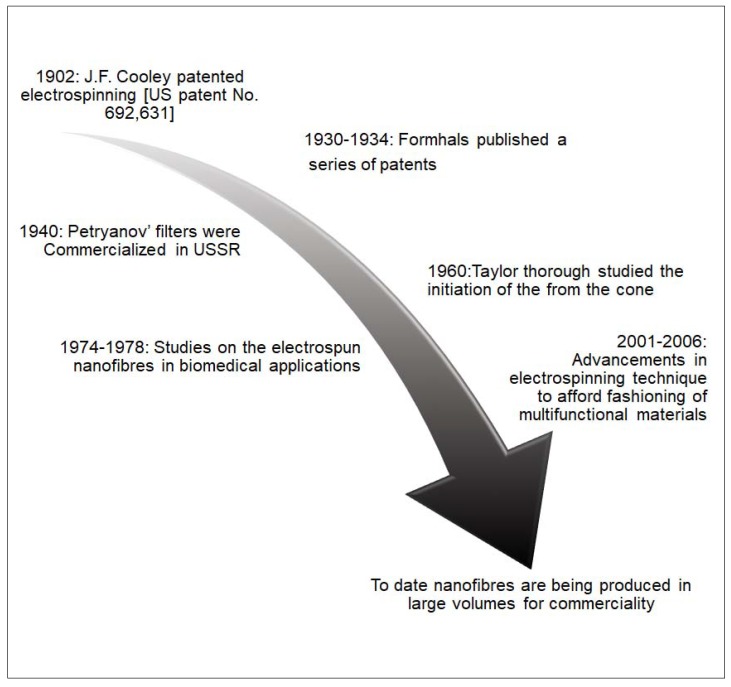
Brief history on the development of electrospinning.

**Figure 3 materials-13-00934-f003:**
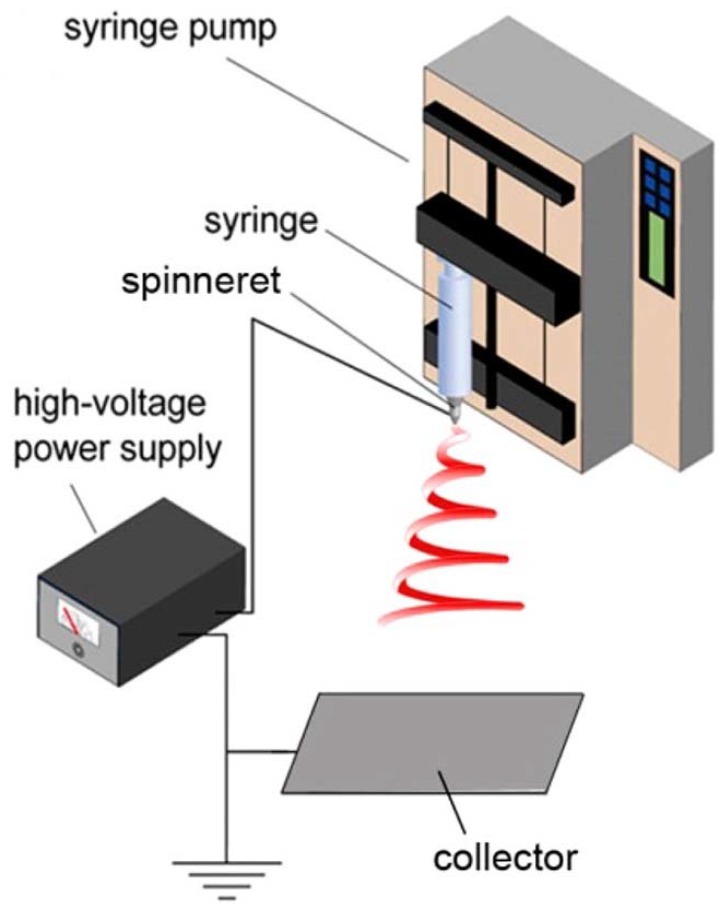
Electrospinning technique setup. Reprinted with permission from [22].

**Figure 4 materials-13-00934-f004:**
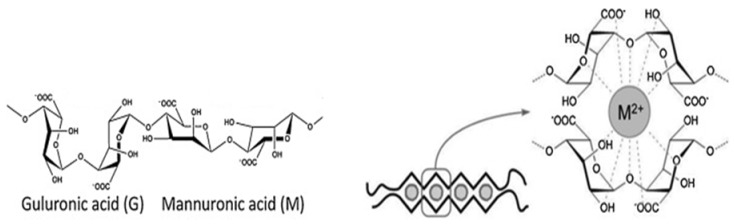
Structure of alginate and “egg-box” model representing alginate organization in the presence of multivalent cations.

**Figure 5 materials-13-00934-f005:**
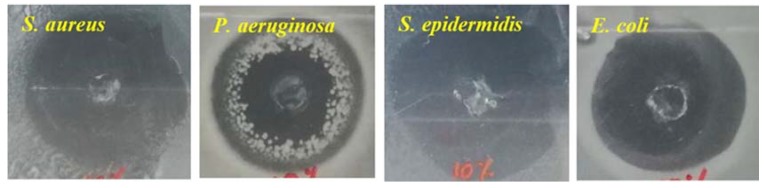
Antimicrobial activity of alginate-based scaffolds with 10% (*w*/*w*) cephalexin (CEF)-HNT. Reprinted with permission from [74]. HNT: halloysite nanotubes and CEF: cephalexin.

**Figure 6 materials-13-00934-f006:**
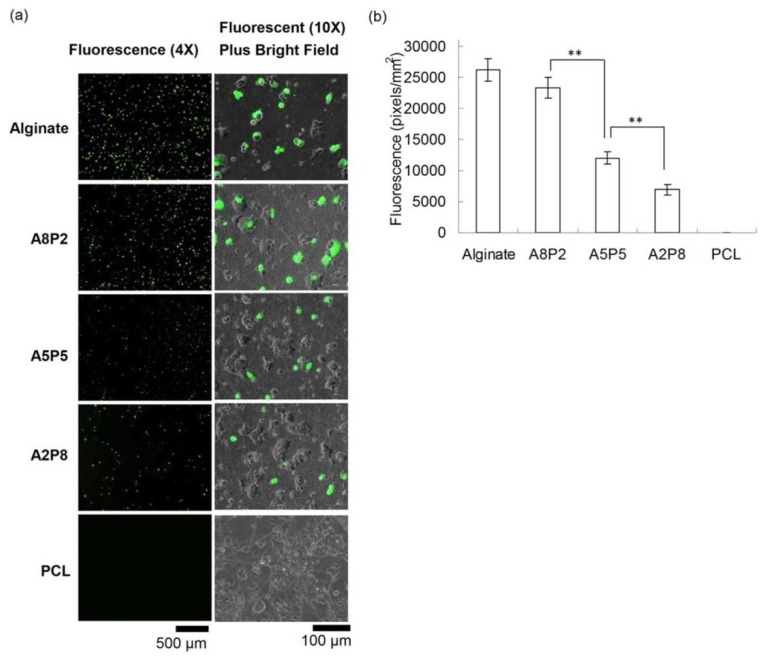
In situ transfection on nanofibers. (**a**) After immobilizing DNA/polyethyleneimine (PEI) nanoparticles, HEK-293T cells were seeded on nanofibers and cultured for 3 days, and the GFP expression from transfected cells was evaluated using fluorescent microscopy. The results suggested the transfection efficiency of composites fibers increased with their alginate ratios. (**b**) The quantification of GFP from fluorescent images also showed the same trend (where, 80% alginate/20% polycaprolactone (PCL), 50% alginate/50% PCL, and 20% alginate/80% PCL, which were denoted as A8P2, A5P5, and A2P8, respectively). Reprinted with permission from [58].

**Figure 7 materials-13-00934-f007:**
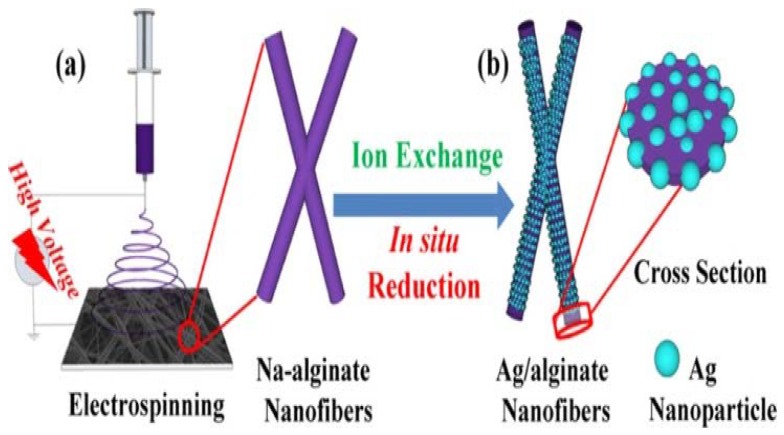
Schematic illustration of Ag–alginate nanofibers fabrication. (**a**) Electrospinning sodium alginate, and (**b**) ion exchange and in situ reduction processes for AgNPs/alginate nanofibers. Reprinted with permission from [85].

**Table 1 materials-13-00934-t001:** Selected studies on the spinnability of alginate.

Type	Additional/Carrier Polymer	Co-Solvent	Surfactant	Technique Type	Optimal Conditions	Highlights	Proposed Application	Refs.
Sodium alginate (SA) 22 KDa	Chitosan as coagulation bath for fiber collection	Glycerol	-	Single needle	Flow rate: 0.1–0.5 mL/hr Tip-to-collector distance (TOC): 7 cm Voltage:13–15 kV	Core–sheath morphology was achieved by electrospinning alginate directly into chitosan coagulation bath, and the fibers having diameter ranging between 600 and 900 nm can be obtained by changing processing parameters	Biomedical	[32]
SA (220 kDa)	Chitosan dissolved in 2.5 wt % acetic acid with 37.5 wt % ethanol co-solvent	Glycerol	-	Single needle	Flow rate: 0.4 mL/hr TOC: 3.5 cm Voltage: 10 kV	Core–sheath morphology coated with collagen/hydroxyapatite (HAp)	Bone tissue engineering	[49]
Sulfated SA (115 kDa)	Poly(vinyl alcohol) (PVA) ^a^	-	-	Single needle	Flow rate: 0.2 mL/hr TOC: 12 cm Voltage: 20 kV	5 mL/hr was demonstrated and obtained fibers with average diameters of 144 ± 21 nm	Biomedical	[43]
SA	Polyethylene oxide (PEO) ^a^	Dimethylformamide	Pluronic F127	Single needle	Flow rate: 0.5 mL/hr TOC: 20 cm Voltage: 25 kV	Cylindrical nanofibers with average diameter of 90 ± 20 nm were obtained and crosslinked with trifluroacetiic acid to afford their stability to more than 14 days in phosphate-buffered saline (PBS) solution at pH value of 7.4	Biomedical	[38]
SA (80–120 kDa, M/G 1.56)	PEO ^a^	Dimethylsulfoxide (DMSO)	Triton X-100	Single needle	Flow rate: 0.8 mL/hr TOC: 17 cm Voltage: 17 kV	Polyelectrlyte complex composed of alginate nanofibers coated with chitosan–AgNPs	Wound dressing	[36]
SA (196 kDa, M/G:1.94)	PEO ^a^	-	Triton X-100	Single needle	Flow rate: 0.5–0.75 mL/hr TOC: 15 cm Voltage: 10–15 kV	Uniform fibers were obtained after addition of Triton X-100 (1 wt %) because of the reduction of surface tension from 55 mN/m to 29 mN/m (70/30 PEO/SA)	Tissue engineering	[31]
SA (196 kDa, M/G: 1.94)	PEO ^a^	-	Pluronic F127			Uniform fibers were obtained after the addition of Pluronic F127 (2 wt %) due to a reduction in surface tension 57 nN/m to 36 mN/m without changing the rheological properties and ionic conductivity of the PEO/SA blend (40/60 SA/PEO)		
SA (37 kDa)	PEO ^a^	-	Pluronic F127			Uniform fibers having diameters of about 150 nm with over 83% alginate in the blend (SA/PEO 8.0/16)	In vivo applications	
SA	PVA ^a^ and ZnO nanoparticles	-	-	Singe needle	Flow rate: 0.1 mL/hr TOC: 5 cm Voltage:17 kV	SA/PVA fibers had diameter ranging between 190 and 240 nm, which increased to 220–360 nm with the inclusion of ZnO nanoparticles into the system	Wound dressing	[29]
SA	PEO ^a^ and Ciprofloxacin hydrochloride (antibiotic)		Triton X-100 or Pluronic F127	Single needle	Flow rate: 0.1–1.0 mL/hr TOC: 15–20 cm Voltage: 6–10 kV	Addition of Triton X-100 led to heterogeneous diameters, while Pluronic resulted in more heterogeneous diametersAntibiotic-loaded fibers had an average of 109–161 nm with loading efficiency of 51%	Wound dressing	[39]
SA (323 kDa, M/G ratio 1.25)	PEO ^a^	Ethanol 10 wt %	Triton X-100 0.8 wt %	Single needle	Flow rate: 0.30 mL/hr TOC: 15–20 cm Voltage: 20–28 kV	Average diameter of 150 nm with mechanical and adsorption capacities were directly depended on the crosslinking agent employed	Dye removal	[59]
SA	Poly (acrylic acid) (PAA) ^a^	-	-	Single needle	Flow rate: 16 µL/hr TOC: 16 cm Voltage: 20 kV Relative humidy (RH): 45 ± 6% Temperature: 35 ± 3 °C	The obtained membrane was thermally crosslinked at 150 °C for 3 h and exhibited an adsorption capacity of 591.7 mg/g	Bioremediation	[60]
SA (80–120 kDa M/G 1.56)	PEO ^a^ and PCL (as co-blended polymer)	Dimethyl sulfoxide (DMSO)	Triton X-100	Dual jet system	Flow rate: 2 µL/min TOC: 20 cm Voltage: 15 kV	Composition with different properties can be achieved by fine tuning the perfusion rate	Gene-delivery	[53,54]
Sulfated alginate	PVA ^a^	-	-	Single needle	Flow rate: 18 mL/hr TOC: 12 cm Voltage: 30 kV	Thermal crosslinking of the electrospun nanofibers improved their integrity and mechanical properties	Tissue engineering	[7]
SA	Poly (acrylic acid) (PAA) ^a^			Single needle	Flow rate: 0.3 mL/hr TOC: 10 cm Voltage: 20 kV	Fibers with diameters of 278.52 ± 64.33 nm and porosity of 42.38	Bone tissue engineering	[32]
SA	PVA ^a^ (MgO NPs as reinforcing agent)	-	-	Single needle	Flow rate: 8–10 µL/min TOC: 10 cm Voltage: 26 kV	Randomly oriented fibers with diameter of 83–230 nm with excellent mechanical properties due to the presence of the NPs	Tissue engineering	[61]
SA	PEO ^a^ (Soy protein as co-blend)	-	-	Single needle	Flow rate: 0.5 mL/min TOC: 15 cm Voltage: 15 kV	Fibers with diameters of 100–300 nm were obtained; however, an increase soy protein content led to beaded fibers	Biomedical	[25]
SA	PVA ^a^ (Honey (acacia as antibiotic)	-	-	Single needle	Flow rate: 0.4 mL/hr TOC: 10 cm Voltage: 15 kV	With honey content being >20% (*v*/*v*) the fibers were more uniform and the membrane exhibited excellent antimicrobial activity against *E. coli* and *S. aureus* without cytotoxicity to NIH/3T3	Wound dressing	[62]
SA	PEO ^a^ (Lavender oil (LO) as antibiotic)	Dimethylformamide (DMF)	Pluronic F127	Single needle	Flow rate: 0.5 mL/hr TOC: 20 cm Voltage: 25 kV	Bead-free fibers with diameters of 93 ± 22 nm were obtained and the membrane exhibited good antibacterial activity against *S. aureus*; meanwhile, inhibited the production of pro-inflammatory cytokines in vitro and in vivo	Wound dressing	[41]
SA	PEO ^a^	DMF	Pluronic F127	Single needle	Flow rate: 0.5 mL/hr TOC: 20 cm Voltage: 25 kV	Using trifluoroacetic acid (TFA), the degradation of electrospun alginate can be adjusted by immersing the fibers at different intervals, which is of significant importance for manufacturing biomedical devices	Regenerative medicine/drug delivery	[63]
Alginate dialdehyde (ADA)	-	Ethanol	-	Single needle	Flow rate: 0.3 mL/hr TOC: 15 cm Voltage: 20–25 kV Temperature: 25 °C Relative Humidity: 30–35%	Oxidation of sodium alginate for 4 h using sodium periodate (NaIO_4_) with the addition of ethanol as co-solvent resulted in bead-free fibers, with the electrospinnability window being improved by incorporating PEO and adipic acid dihydrazide (AAD, crosslinking agent)	Biomedical	[42]

^a^ carrier polymer.

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
