# Peer review of "Electrospun Alginate Nanofibers Toward Various Applications: A Review"

_materials, 2020, doi:10.3390/ma13040934_

Round 1

Reviewer 1 Report

In this review paper the authors described the recent use of electrospun alginate nanofibres in various fields, such as: bioremediation, scaffolds for skin tissue engineering, drug delivery and sensors. I must state that this is a very weak review paper and I recommend a rejection of the paper. This is always hard to do considering the efforts made by the authors but it is also one of the reasons for the peer review process.

My comments about the manuscript are as follows:

- The Introduction is very poor.

- The ideas of the MS seem to be a little spread out through some parts of the paper, going from electrospinnability of alginate to applications. Try to make transition areas flow more, so that the reader can understand why you are comparing two ideas together that alone seem independent.

- I think this paper seems to be based on several good papers found in the literature. However, I noticed that the authors have ignored important papers in the field. Make sure to explain why these studies were added and the significance of each one with relevance to the paper. Sometimes I was lost as to why a study was referenced and what it was being compared to.

- There are many review papers (DOI: 10.1016/j.carbpol.2019.115419, DOI: 10.1080/15583720802022182, DOI: 10.1089/ten.2006.12.1197 to name a few) on this topic. Thus, most of the aspects described by the authors in their manuscript are cover in the above literatures. The introduction would be improved by explicitly stating the novelty that this review will address.

- How is it possible to propose a review article in which the corresponding authors is only cited very few times? It does not make sense.

- Lastly, the presentation (including the figures) is VERY poor and substantial grammatical corrections are required to make the manuscript readable.

Author Response

Reviewer 1

Comments and Suggestions for Authors

In this review paper the authors described the recent use of electrospun alginate nanofibres in various fields, such as: bioremediation, scaffolds for skin tissue engineering, drug delivery and sensors. I must state that this is a very weak review paper and I recommend a rejection of the paper. This is always hard to do considering the efforts made by the authors but it is also one of the reasons for the peer review process.

My comments about the manuscript are as follows:

Response: Thank you for such valuable comments to improve the quality of the manuscript; we have included more information such that the manuscript is of good quality.

- The Introduction is very poor.

Response: we have included some new information to improve it.

- The ideas of the MS seem to be a little spread out through some parts of the paper, going from electrospinnability of alginate to applications. Try to make transition areas flow more, so that the reader can understand why you are comparing two ideas together that alone seem independent.

Response: we have included some new information to make the discussion under question to be better and have a flow throughout the manuscript.

- I think this paper seems to be based on several good papers found in the literature. However, I noticed that the authors have ignored important papers in the field. Make sure to explain why these studies were added and the significance of each one with relevance to the paper. Sometimes I was lost as to why a study was referenced and what it was being compared to.

Response: we have cited more papers in order to improve the manuscript

- There are many review papers (DOI: 10.1016/j.carbpol.2019.115419, DOI: 10.1080/15583720802022182, DOI: 10.1089/ten.2006.12.1197 to name a few) on this topic. Thus, most of the aspects described by the authors in their manuscript are cover in the above literatures. The introduction would be improved by explicitly stating the novelty that this review will address.

Response: we have added some information to make our introduction to flow as requested

- How is it possible to propose a review article in which the corresponding author is only cited very few times? It does not make sense.

Response: It is true we have published few papers based on the electrospinnability of alghinate and its application in wastewater treatment and biomedical applications, but it took us more than 6 years on the project which gave us a quite experience on the electrospinnability of alginate as well as its applicability. Again we are still busy working on this kind of project where we look at the applicability of the electrospun natural polymers which we think it’s a worthwhile to publish review to give insight for other researchers who want to pursue this kind of field. 

- Lastly, the presentation (including the figures) is VERY poor and substantial grammatical corrections are required to make the manuscript readable.

Response: we improved and fixed all grammatical errors

Reviewer 2 Report

Dear Editor,

The manuscript submitted by Mokhena et al. and entitled: “Electrospun alginate nanofibers towards various applications: a review” is an interesting study which deals with the development of alginate nanofibers for industrial applications. This manuscript can be accepted after major revisions. Please to see my comments below.

Comments:

In introduction part, authors have made lot of statements without backed it up with last good references in this field. Please to add new references in the revised manuscript. In the conclusion part, the aspects of novelty and the mains futures applications should be more underlined by comparison with actual industrial applications.

General comment:

In the revised manuscript, the authors need to pay more attention to grammatical construction of sentences and spelling of sentences. The quality of figures must be improved in the revised manuscript.

Author Response

Reviewer 2

Comments and Suggestions for Authors

Dear Editor,

The manuscript submitted by Mokhena et al. and entitled: “Electrospun alginate nanofibers towards various applications: a review” is an interesting study which deals with the development of alginate nanofibers for industrial applications. This manuscript can be accepted after major revisions. Please to see my comments below.

Comments:

In introduction part, authors have made lot of statements without backed it up with last good references in this field. Please to add new references in the revised manuscript. In the conclusion part, the aspects of novelty and the mains futures applications should be more underlined by comparison with actual industrial applications.

Response: We would like to thank the reviewer for such valuable comments in order to improve the quality of the manuscript. We have included the citation to back up all the statements throughout the manuscript.

General comment:

In the revised manuscript, the authors need to pay more attention to grammatical construction of sentences and spelling of sentences. The quality of figures must be improved in the revised manuscript. 

Response: we replaced the figures with high quality ones. We fixed all grammatical errors as indicated by the reviewer.

Reviewer 3 Report

Authors have reviewed the studies based on electrospun alginate nanofibres for various applications. This review article may be interesting to researchers and research professionals for future studies.  In addition, various systems and polymers have been presented for better electrospinning ability of alginate-based solutions. However, this review does not provide comprehensive application-based information and just provided superficial review. This manuscript needs to be re-formatted for better presentation of the point of specific study.

Introduction section should be elaborated for more effective background of alginate and their advantages and disadvantages. Also, authors should check carefully the cited Ref in lines 44-46. There is a large number of publications published between 2009 and 2019 based on electrospinning of alginate for various applications (more than 350 articles), but authors did not include comprehensive literature on published articles in this review. Authors have used nearly 60 articles and around 20 articles for applications part in this review. In my opinion, authors should focus on application of electrospun alginate nanofibres in a comprehensive manner. In Table 1, most of the data is based on biomedical applications; especially tissue engineering and only one article is for bioremediation (i.e. dye removal). Is this review specific to biomedical applications or all industrial and biomedical applications, as claimed in the title of this review?. If this is focused on biomedical applications, then there are already articles published on it, for example: Biopolymers for biomedical and pharmaceutical applications: Recent advances and overview of alginate electrospinning. Nanomaterials9(3), 2019, 404 and Fabrication Challenges and trends in biomedical applications of alginate electrospun nanofibers. Carbohydrate polymers, 2019, 115419. Authors should clarify how this review is different and effective for researchers for future studies. Authors should elaborate application section for better representation of this material. In addition, good Figures prepared by authors of this review or from published articles for actual concept of studies should be provided. Others Figures are also not in good resolution and please provide better quality of all figures. Future trend and conclusions should be re-worded in terms of limitations of alginate electrospinning and their advantages for future perspectives.

Author Response

Reviewer 3

Comments and Suggestions for Authors

Authors have reviewed the studies based on electrospun alginate nanofibres for various applications. This review article may be interesting to researchers and research professionals for future studies.  In addition, various systems and polymers have been presented for better electrospinning ability of alginate-based solutions. However, this review does not provide comprehensive application-based information and just provided superficial review. This manuscript needs to be re-formatted for better presentation of the point of specific study.

Response: Thank for such valuable comments in order to improve the quality of our manuscript

Introduction section should be elaborated for more effective background of alginate and their advantages and disadvantages. Also, authors should check carefully the cited Ref in lines 44-46. There is a large number of publications published between 2009 and 2019 based on electrospinning of alginate for various applications (more than 350 articles), but authors did not include comprehensive literature on published articles in this review. Authors have used nearly 60 articles and around 20 articles for applications part in this review. In my opinion, authors should focus on application of electrospun alginate nanofibres in a comprehensive manner. In Table 1, most of the data is based on biomedical applications; especially tissue engineering and only one article is for bioremediation (i.e. dye removal). Is this review specific to biomedical applications or all industrial and biomedical applications, as claimed in the title of this review?. If this is focused on biomedical applications, then there are already articles published on it, for example: Biopolymers for biomedical and pharmaceutical applications: Recent advances and overview of alginate electrospinning. Nanomaterials9(3), 2019, 404 and Fabrication Challenges and trends in biomedical applications of alginate electrospun nanofibers. Carbohydrate polymers, 2019, 115419. Authors should clarify how this review is different and effective for researchers for future studies. Authors should elaborate application section for better representation of this material. In addition, good Figures prepared by authors of this review or from published articles for actual concept of studies should be provided. Others Figures are also not in good resolution and please provide better quality of all figures. Future trend and conclusions should be re-worded in terms of limitations of alginate electrospinning and their advantages for future perspectives.

Response: we have improved the manuscript as requested by the reviewer.

Reviewer 4 Report

This is a well-written, well-researched article on the use of alginate in electrospinning applications. 

The authors need to address the following points to make the review more streamlined:

1) In the bioremediation section,

a) the pore size of alginate membranes is mentioned.  Are micro-porous alginates also used and what are the ranges of pore sizes?

b) Alginate is extensively itself doe not have antimicrobial properties so the legend of Figure 5 need to get addressed.

c) If possible create subsections:

a) Filtration

b) biodegradation

2) The biomedical applications need to be expanded in subsections. The following are proposed but the authors can create alternative ones:

a) Wound Healing

Bioengineering 2018, 5(1), 9; https://doi.org/10.3390/bioengineering5010009

b) Materials/Tissue Engineering /Immobilization

Contents from other applications should be moved to this section.

3) Some physical properties are alluded to with no references.and transitions are not explained. This should be reviewed throughout the article.

i,e What is the Young's modulus of human skin? This varies age and disease states.... and where is the Young's modulus of alginate?

4) The applications section should be renamed to "sensors and energy" with references to explain the properties alluded to (i.e lines 147-150).

5) The conclusion should be rewritten in light of the restructuring of some of the sections.

6) The resolution of Figures needs to be improved.

7) Line 143: chiton should be replaced by chitosan.

Author Response

Reviewer 4

Comments and Suggestions for Authors

This is a well-written, well-researched article on the use of alginate in electrospinning applications. 

Response: Thank for your valuable responses to improve our manuscript

The authors need to address the following points to make the review more streamlined:

1) In the bioremediation section,

a) the pore size of alginate membranes is mentioned.  Are micro-porous alginates also used and what are the ranges of pore sizes?

Response: to the best of our knowledge we haven’t come across studies based on the microporous alginate-nanofibrous membranes, hence not discussed

b) Alginate is extensively itself does not have antimicrobial properties so the legend of Figure 5 need to get addressed. c) If possible create subsections: a) Filtration b) biodegradation

Response: We did create subsections as requested by the reviewer.

2) The biomedical applications need to be expanded in subsections. The following are proposed but the authors can create alternative ones:

a) Wound Healing

Bioengineering 20185(1), 9; https://doi.org/10.3390/bioengineering5010009

Response: we subdivided the biomedical application section, and the proposed reference was also used to improve the manuscript

b) Materials/Tissue Engineering /Immobilization

Contents from other applications should be moved to this section.

Response: we have moved all sections accordingly

3) Some physical properties are alluded to with no references and transitions are not explained. This should be reviewed throughout the article.

Response: This has been addressed

i,e What is the Young's modulus of human skin? This varies age and disease states.... and where is the Young's modulus of alginate?

Response: It is true that the human skin properties can change depending on several factors, such as age, infections, etc. but we alluded is that controlling the parameters of electrospinning and modifying alginate renders opportunity to control the resulting mechanical properties of the membrane such that it can afford its application in the proposed wound dressing

4) The applications section should be renamed to "sensors and energy" with references to explain the properties alluded to (i.e., lines 147-150).

Response: We have included the references to backup the alluded properties

5) The conclusion should be rewritten in light of the restructuring of some of the sections.

Response: This has been addressed

6) The resolution of Figures needs to be improved.

Response: The Figures were removed and replaced with high quality ones

7) Line 143: chiton should be replaced by chitosan.

Response: This has been addressed

Round 2

Reviewer 1 Report

This reviewer believes this topic could be of interest of Materials. The authors adequately responded to most of the questions in the resubmitted manuscript. However, while they have improved the quality of MS, the paper still fails to address what has not been addressed elsewhere. The novelty of this work was not adequately clarified and covered. This reviewer feels that this is an article that covers what is already well covered by other review articles. The authors should list and compare other reviews on the same or similar topic that have been published in the previous five years.

Author Response

Reviewer 1

Response: thank you for such valuable comments, to the best of our knowledge there are only two reviews that are closely related to our manuscript as discussed below. Other reviews discussed electrospun bio-based polymers in general, such as alginate, chitosan, hyaluronic, cellulose etc. towards various applications.

MA Taemeh, A Shiravandi, MA Korayem, H Daemi. Fabrication challenges and trends in biomedical applications of alginate electrospun nanofibers. Carbohydrate Polymers 2020, 228:115419

In this review the authors looked into the difficulties associated with electrospinning alginate and proposed techniques to facilitate its spinnability.  The authors concluded that wound dressing is more ideal application for electrospun alginate nanofibers with the potential in some applications such as drug delivery and tissue engineering. In addition, authors summarized the available literature on the electrospinnability of alginate and did not give any future recommendations in order to overcome some issues associated with electrospinnability and applications of alginate in applications other than biomedical.

J Wróblewska-Krepsztul, T Rydzkowski, I Michalska-Pożoga I, VK Thakur . Biopolymers for Biomedical and Pharmaceutical Applications: Recent Advances and Overview of Alginate Electrospinning. Nanomaterials 2019; 9.

The authors summarized the applicability of the electrospun alginate in biomedical and packaging. The review compared synthetic and bio-based polymers’ properties towards biomedical and packaging applications with special emphasis on alginate for biobased polymers. The authors concluded that there are very limited studies on the electrospun alginate nanofibres on packaging in order to realize their potential.

In our work we summarized the available literature on the electrospinnability of alginate over the past ten years. We also summarized applications of the electrospun alginate nanofibers other than the classic applications, viz biomedical. We also give recommendations associated with electrospinnability of alginate as well as its potential in other applications which is often overlooked despite the attractive attributes of alginate. Our aim was to give the readers with cited literature a reference point along with roadmaps for future research efforts required considering much interest of electrospinning alginate in the past ten years.

Reviewer 2 Report

Dear Editor,

Authors well revised the manuscript according to reviewer's comments.

Consequently, this paper could be accepted in your journal.

Best regards,

Author Response

Response: Thank you

Reviewer 3 Report

In my opinion, this manuscript now can be accepted for publication. 

Author Response

Response: Thank you

Reviewer 4 Report

Thank you for making the comprehensive changes.

Author Response

Response: Thank you

Round 3

Reviewer 1 Report

The manuscript can be accepted in present form.